# Effect of Curing Modes on the Mechanical Properties of Commercial Dental Resin-Based Composites: Comparison between Different LEDs and Microwave Units

**DOI:** 10.3390/polym14194020

**Published:** 2022-09-26

**Authors:** Alef Vermudt, Milton Carlos Kuga, João Felipe Besegato, Eliane Cristina Gulin de Oliveira, Thaís Piragine Leandrin, Marcus Vinicius Reis Só, João Carlos Silos Moraes, Jefferson Ricardo Pereira

**Affiliations:** 1Postgraduate Program in Health Sciences, University of Southern Santa Catarina–Unisul, 787 José Acácio Moreira Tubarão Street, Tubarão 88704-900, Brazil; 2Department of Restorative Dentistry, School of Dentistry, Araraquara, São Paulo State University (UNESP), 1680 Humaitá Street, Araraquara 14801-903, Brazil; 3Department of Endodontics, School of Dentistry, University Federal of Rio Grande do Sul (UFRGS), Farroupilha, Porto Alegre 90010-150, Brazil; 4Department of Physics and Chemistry, School of Natural Sciences and Engineering, Ilha Solteira, São Paulo State University (UNESP), 56 Brasil Avenue, Ilha Solteira 15385-000, Brazil

**Keywords:** resin composites, curing, flexure strength, LED source, degree of conversion

## Abstract

Resin-based composites (RBCs) have transformed restorative dentistry and its procedures. However, the characteristics of RBCs have been modified over the years to enhance the physical and chemical properties of the materials. This context raises the need for studies that evaluate whether the properties of the RBCs that are commercially available are clinically adequate with different curing modes. This study aimed to evaluate the mechanical behavior of commercial RBCs after undergoing different curing modes. Twenty-three RBCs of different classes were evaluated. For curing the specimens, a microwave (BMS45, Brastemp) (for 3 min at 450 W) and three LED units were used: an Emitter A Fit (Schuster (second generation)) (light-curing for 15 s with an irradiance of 1250 mW/cm^2^), VALO (Ultradent (third generation)) (light-curing for 15 s with an irradiance of 1100 mW/cm^2^), and Emitter Now Duo (Schuster (second generation)) (light-curing for 15 s with an irradiance of 1100 mW/cm^2^). A total of 670 RBC specimens of 8 mm in diameter and 1 mm in depth were obtained. Afterward, a biaxial flexure strength test was performed until the failure of the specimens, using a universal testing machine set at a speed of 0.5 mm/min. The same specimens were subjected to infrared spectroscopy for evaluating the degree of conversion. Tukey’s test was used for multiple comparisons at a significance level of 5%. The light-curing mode did not affect the flexure strength of the RBCs (*p* > 0.05), but the type and shade of RBCs did so (*p* < 0.05). In conclusion, the type of RBC directly interferes with the mechanical behavior of the material. However, the curing modes within the same RBC did not change the mechanical properties.

## 1. Introduction

For many years, dental cavities have been filled with different materials to restore lost dental structures and reestablish the form and function of teeth. With technical and scientific developments, restorative materials have improved from amalgam and acrylic resins to resin-based composites (RBCs) [1,2].

RBCs are dental materials based on organic and inorganic resinous compounds. The organic matrix is composed of resinous monomers, silane coupling agents, and initiator molecules. The inorganic content is characterized by filler particles of different compositions (silica, quartz, and glass), sizes, and shapes. The amount of each component of an RBC can vary according to the manufacturer. Thus, the great diversity in the materials’ compositions challenges dental clinicians to make the most appropriate clinical choice in terms of material, trademark, shade, and chemical and physical properties of the RBC [1,2,3].

Over the years, the characteristics of RBCs were modified in an attempt to improve their physicochemical properties. Nevertheless, laboratory and clinical studies that evaluate these novel types of materials are crucial to deeply exploring and predicting their clinical behavior [1,3].

The physicochemical properties of RBCs are directly related to the degree of conversion of monomers into polymers after curing. In the case of dental RBCs, light units are widely used for curing since they emit wavelengths in the visible spectrum that are capable of converting the resinous monomers into complex polymers. This process is known as photopolymerization. Thus, the greater the conversion of monomers into polymers, the greater the final resistance of the RBC [3].

The polymerization reaction is:

r

R-C-O-O-C-R (iniciator) + activator = R-C-O* (free radical (FR))

R-C-O* + C=C = R-C-O-C-C* (FR)

R-C-O-C-C-C-C* + C=C = R-C-OC-C-C-C-C-C*

R-C-O-C-C* (FR) + C*-C-O-C-R (FR) = R-C-O-C-C-C-C-O-C-R (polymer)

An insufficient degree of conversion (the number of monomers converted to polymers) can result in poor mechanical and physical properties that trigger several undesirable effects in dental restorations, such as microcracks, fracture, marginal staining, and microleakage [4,5,6]. Nevertheless, there are certain clinical strategies that can diminish or avoid these undesirable effects, which involve the mode and unit used for curing the RBCs [4,5,6].

The degree of conversion is influenced by the way that light is transmitted through the composite resin. The intensity and time in which the light hits the material may be the reason why composite resins may lose strength and fracture early [7].

The current composite resins are photopolymerizable systems activated by visible light, wherein a photoinitiator is required to react and generate free radicals and thereby initiate the chain polymerization of the compound. Camphorquinone is the most frequently used photoinitiator; when it is irradiated by light of the appropriate wavelength, which would be close to 465 nm, the monomer groups react, and polymers are formed. It is present in most composite resins because all photopolymerization units on the market can reach the necessary wavelength to excite it and can then start the polymerization process [8].

The presence of camphorquinone in the composite resin, due to its strong yellowish color, can affect the aesthetics of these materials when it is necessary to use white and transparent materials, so researchers considered replacement with or the inclusion of alternative photoinitiators that did not influence the final aesthetics of the resin restoration [9,10].

Among all the photoinitiators used to replace camphorquinone, 1-phenyl-1; 2-propanedione (PPD), diphenyl oxide (2,4,6-trimethylbenzoyl) phosphine (Lucirin TPO), and bis-alkyl phosphine oxide irgacure (BAPO) differ mainly by having white/transparent or slightly yellowish colors, which allows the possibility of achieving a better aesthetic. However, these components require light with wavelengths of around 398 nm for PPD, 400 nm for BAPO, and 380 nm for Lucirin TPO, which, in fact, can become a problem since most light-curing equipment reaches a wavelength waveform from 420 to 480 nm [11,12].

The dental manufacturers developed a great variety of RBCs. However, many of the RBCs have not been tested under different conditions using scientific research. Moreover, the effect of different curing modes and units on the mechanical and physical properties of commercially available RBCs is still uncertain. Thus, this study aimed to evaluate how RBCs behave after the employment of curing modes using different curing units, in terms of flexure strength. The null hypothesis that was tested was that the curing modes and units did not influence the biaxial flexure strength of RBCs.

## 2. Materials and Methods

Fourteen commercial RBCs were selected for this study. The shade that was standardized was A2, based on the VITA classical scale (Wilcos, Rio de Janeiro, RJ, Brazil), except for nine RBCs that also used different classifications of shade. Table 1 shows the trademark, classification, and manufacturer of each RBC tested.

For curing the RBC specimens, three LED units were used: Emitter A Fit (Schuster, Santa Maria, RS, Brazil (second generation)), VALO (Ultradent Products Inc., South Jordan, UT, USA (third generation)), and Emitter Now Duo (Schuster, Santa Maria, RS, Brazil (second generation)). A microwave (BMS45, Brastemp, São Paulo, SP, Brazil) was also used.

After the selection of materials and curing units, 670 RBC specimens of 8 mm in diameter and 1 mm in depth were obtained (Table 2) using an acrylic matrix (Figure 1).

The RBC specimens were then divided into seven groups, according to the curing modes—Group 1: Valo + Shade A2; Group 2: Emitter A Fit + Shade A2; Group 3: Emitter Now Duo + Shade A2; Group 4: Emitter A fit + Shade A2 + microwave; Group 5: Valo + Effects Shades; Group 6: Emitter A Fit + Effects Shades; Group 7: Emitter Now Duo + Effects Shades—as described in Table 2. The curing modes used were: VALO (light-curing for 15 s with an irradiance of 1100 mW/cm^2^); Emitter A Fit (light-curing for 15 s with an irradiance of 1250 mW/cm^2^); Emitter Now Duo (light-curing for 15 s with an irradiance of 1250 mW/cm^2^); and an additional curing with a microwave (BMS45, Brastemp) for 3 min at 450 W. The VALO and Emitter Now Duo are third-generation photopolymerization units; they have blue, violet, and ultraviolet light that achieves photoinitiation. while Emitter Fit is a second-generation unit that has only a blue light.

The VALO and Emitter Now Duo are LED third-generation units with wavelengths of 395–480 nm and 385–515 nm, respectively. On the other hand, the Emitter A Fit is an LED second-generation unit with a wavelength of 420–480 nm. The irradiance of each LED unit was checked using a digital radiometer (RD-7, ECEL) that captures visible light, with wavelengths of between 400 and 500 nm and irradiance from 0 to 1270 mW/cm^2^, with an accuracy of ±5%, as in Table 3.

The RBC specimens were evaluated with a stereomicroscope (Stemi DV4, Zeiss, Oberkochen, Germany) to verify structural defects and avoid variability among them. Those specimens with cracks, staining, and missing material were excluded. Then, the included RBC specimens were stored in a dark, clean, dry environment.

### 2.1. Biaxial Flexure Strength Test

The biaxial flexure strength (BFS) test was performed as described by Rueggeberg et al. [11], using a universal testing machine (Kratos, São Paulo, SP, Brazil). Each RBC disk was placed in a custom-made matrix with a circumference of 1 mm. Then, the disk was centrally loaded with a ball-end plunger (0.5 mm in diameter) at the center of the testing apparatus, at a cross-head speed of 0.5 mm/min until the failure of the specimen (Figure 2). After that, the BFS was recorded.

### 2.2. Degree of Conversion

To assess the degree of conversion, specimens were prepared using a portion of uncured RBC, around 2 mm^2^, which was placed and compressed between two microscope slides. The degree of conversion (DC) was evaluated using the absorption band at 4743 cm^−1^ (Figure 3) and can be attributed to aliphatic =CH_2_ bonds, which progressively decrease during the polymerization reaction. The absorption band at 4585 cm^−1^, assigned to aromatic =CH_2_ bonds, was used as an internal standard for normalization since its intensity is unaltered during the polymerization reaction. The percentage of reacted aliphatic =CH_2_ bonds (=DC) was obtained by the equation:$$\mathrm{DC}\%=\left(1-\frac{{\left(\frac{I_{4743}}{I_{4585}}\right)}_{\mathrm{cured}}}{{\left(\frac{I_{4743}}{I_{4585}}\right)}_{\mathrm{uncured}}}\right)\times 100$$

The NIR infrared spectra of cured and uncured films were obtained from 96 scans at a resolution of 4 cm^−1^ on an FTIR spectrometer (Nexus 670 from Nicolet, Madison, WI, USA). The RBC films were cured according to the curing unit: VALO–15 s at 1100 mW/cm^2^; Emitter A Fit–15 s at 1250 mW/cm^2^; Emitter Now Duo–15 s at 1250 mW/cm^2^. The spectra of the cured films were obtained 90 s after they were irradiated. During light-curing, the LED unit tip was placed to cover and almost touch the entire surface of the RBC specimen. Figure 3 shows an example of how those RCBs changed in terms of characterization (via FTIR) before and after activation.

### 2.3. Statistical Analysis

The data from the BFS test were tabulated using the Excel program (v16.0)(Microsoft, Washington, DC, USA). An ANOVA test was used to establish the statistical variance (*p* < 0.05). Tukey’s test was used for multiple comparisons at a significance level of 5%, using the SPSS statistics software (v16.0)(IBM, New York, NY, USA).

## 3. Results

The results of the BFS test are shown in Table 4, Table 5, Table 6 and Table 7, while the results from the degree of conversion are shown in Table 8.

When comparing second- and third-generation LED curing units within the same shade of RBC (A2), no statistically significant differences were found (*p* > 0.05). However, comparing different RBCs cured with the same unit, statistically significant differences were found (*p* < 0.05). Additional curing using a microwave (Table 5) showed no statistically significant differences when compared to the second- and third-generation LED curing units.

Table 6 and Table 7 display the comparisons of RBCs by effects shade according to the different LED units (second and third generations). No differences were found within the same RBC (*p* > 0.05), but when comparing the same LED unit, the type of RBC affected the BFS (*p* < 0.05). Table 8 shows the degree of conversion from different RBCs. The greater the degree of conversion (monomers converted into polymers), the more polymerized the composite resin [3].

## 4. Discussion

The properties of light-cured RBCs can be altered depending on the curing unit used. This study aimed to explore whether the curing mode affects the flexure strength of commercial RBCs using an ANOVA and Tukey test for multiple comparisons. The fracture of RBC is a frequent cause of failure of RBC restorations since insufficient polymerization can lead to microcracks or the incorporation of voids within the RBC during the filling process. Based on our results, the null hypothesis was partially accepted, since the curing unit did not influence the properties of the same RBC. However, differences were found when different RBCs were cured with the same unit.

The classification of RBCs varies according to the viscosity, inorganic filler content, shade, and amount of organic matrix. Our results showed that these characteristics can affect the physical properties of RBC in terms of flexure strength. Grandio and GrandioSO are hybrid composites with 87% and 89% of inorganic filler, respectively, which may explain the high BFS values. Thus, the lower percentage of inorganic filler for Palfique (71%) and Empress (77%) may explain the lower BFS values. However, the same did not occur with Z350, a nanofilled RBC with 78% of inorganic filler, which showed high BFS values. The inorganic nano-agglomerates in the composition of Z350 may improve their resistance and justify the enhanced BFS. In this way, the composition and characteristics of the RBCs explain the differences observed among RBCs cured with the same unit (Table 5 and Table 7), which is in accordance with previous studies [12,13,14,15].

Beun et al. [12] compared the mechanical properties of nanofilled, hybrid, and microfilled RBCs. As demonstrated in our study, nanofilled RBCs such as Palfique showed lower values compared to hybrid ones, such as Grandio and GrandioSO, regardless of the curing unit used. The authors also showed a statistically significant difference when comparing hybrid and nanofilled RBCs. The smaller the amount of resin matrix, which is associated with the filler size, the greater the flexure strength over time, due to the decrease in the degradation of the resin matrix and the bonds between the matrix and the filler particles [12,13,15].

RBC is one of the most successful filling materials for dental restorations due to its mechanical and optical properties, which are similar to the tooth structure. The variations in those properties in each RBC alter the light-curing behavior, which changes the resistance of the RBC, according to a review by Kowalska et al. [16]. However, in our study, the properties of the same RBC were not affected by different curing modes [16,17].

The different compositions of RBCs in relation to the organic matrix, and the amount and size of inorganic filler, affect the mechanical properties of these materials, which may explain our results. Therefore, the material’s properties are composition-dependent, to obtain adequate light-curing and resistance, corroborating our results, where light-curing with different units that emit different irradiances did not show significant differences [3,18,19].

The curing units tested in this study showed no significant differences in the BFS of RBC. This fact corroborated previous studies and highlighted that the changes in BFS of light-cured RBC are dependent on the size of the increment, exposure time, and distance from the light source. In this study, it can be seen that the use of second- or third-generation LED units did not alter the flexure strength of a 1-millimeter-thickness of the material, perhaps due to the fact that the light tip was not moved away from the material’s surface, improving the light penetration. Furthermore, this can be a disadvantage as it can decrease the degree of conversion and increase the shrinkage of the RBC [20,21,22,23].

The light penetration into an RBC increment can be altered according to its thickness and shade [24]. In this study, to evaluate the degree of conversion, we used a material thickness that varied from 170 to 200 μm, which may have contributed to obtaining similar results, regardless of the composite resin [25].

The Vittra of shade E, Bleach, was the only material that showed different values of the degree of conversion. A difference of 18% was observed when comparing third-generation LEDs and of almost 35% when comparing VALO and Emitter A Fit equipment. More detailed information regarding the material’s composition must be provided by the manufacturer to explain this result since the shade can change the RBC composition [26].

A literature review regarding the initiator molecules of RBC was recently carried out [11]. In this article, the authors cited alternative photoinitiators (PPD, Lucirin TPO, and BAPO) and confirm that they are activated at wavelengths below 420 nm. Theoretically, a second-generation LED unit would not be able to activate them since wavelengths ranging from 420 nm to 480 nm are emitted. In this research, the degree of conversion of RBCs was very similar within the same curing unit, except for the Vittra E-Bleach resin, which had a degree of conversion that was very low when using a second-generation LED. The explanation of this result requires more detailed research regarding its chemical components, since the manufacturer does not specifically describe each component. In this context, in 2006, Neumann et al. highlighted the importance and duty of manufacturers to provide the absorption profile of RBC and the spectral emission range of curing units [27]. Moreover, the use of light-curing units did not change the properties of RBCs, but the light exposure time and the distance of the light tip from the material’s surface showed significant differences [11,24].

It is important to note that most of the RBCs tested showed a degree of conversion below 50%, which must be a concern for dental clinicians since low polymerization (polymerization should achieve, at minimum, a 60% to 65% degree of conversion) can generate several undesirable effects in terms of the longevity of RBC restorations [27]. The sub-polymerized composite resin will generate undesirable effects on dental restorations, such as microcracks, fracture, marginal staining, and microleakage [5,6,28]. Thus, further studies are needed to evaluate different curing modes, focusing on exposure time and distance from the light source.

There is also, as demonstrated in the literature, the possibility of increasing the degree of conversion of a composite resin using a microwave. This would make a restoration, made indirectly (veneer, onlay, inlay, or overlay), better compared to another one polymerized directly in the mouth, thus increasing its resistance [29,30,31]. Instead of this possibility, this study showed no statistically significant differences among the groups using or not using the additional energy from microwaves. This can be explained by observing the distance of the light tip from the material’s surface, which can show a significant decrease in the degree of conversion and an increase in the shrinkage of the RBC [11,24]; because of that, the use of the microwave would be not enough to improve the resistance of RBC.

Within the limitations of this in vitro study, the mechanical behavior of RBCs was similar when the same curing unit was used to polymerize the specimens. However, it is questionable that the way in which the polymerization was carried out could be a contributory factor to these results (the source of light was toughening up the composite resin). There are alternatives to overcome these limitations, such as changing the distance of the light source from the composite resin or the use of high-intensity LED sources. Novel studies are needed to investigate whether these factors affect the resistance of RBCs. Since the microwave is a method used outside the mouth for curing indirect restorations, other dental laboratory-curing units might be tested instead, with LED units; this could be considered a limitation when the treatment was a direct restoration.

## 5. Conclusions

Within the limitations of this study, some conclusions can be drawn:The type of resin-based composite showed different flexure strength values;The curing mode did not affect the flexure strength of resin-based composites;The shade of resin-based composites did not interfere with the flexure strength;The degree of conversion of resin-based composites was not affected by the curing unit used, except for Vittra E Bleach.

## Figures and Tables

**Figure 1 polymers-14-04020-f001:**
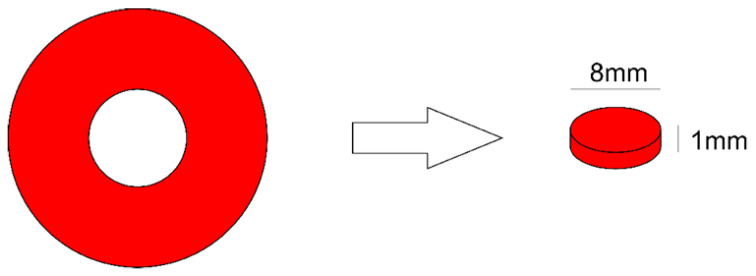
Scheme of the Teflon matrix used to obtain the discs of resin-based composites.

**Figure 2 polymers-14-04020-f002:**
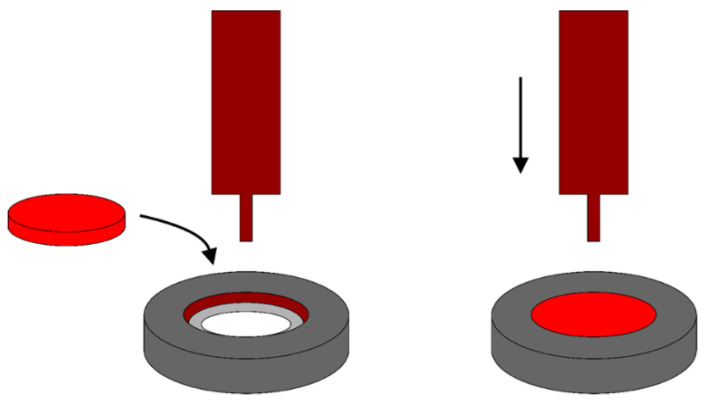
Scheme of the biaxial flexure strength test.

**Figure 3 polymers-14-04020-f003:**
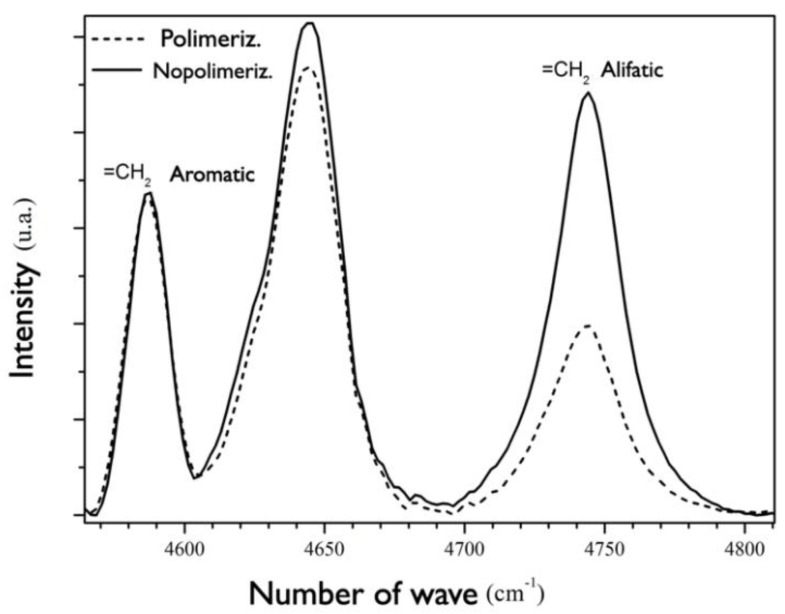
Absorption spectrum of resin−based composites before (SPol) and after (Pol) polymerization.

**Table 1 polymers-14-04020-t001:** Resin-based composites (RBC) used in this study.

RBC Trademark	Classification	Manufacturer
Charisma	Microhybrid	Heraeus Kulzer
Z100	Microhybrid	3M ESPE
Z250	Microhybrid	3M ESPE
Polofil Supra	Microhybrid	VOCO
Empress Direct	Nanohybrid	Ivoclar Vivadent
Harmonize	Nanohybrid	Kerr
Herculite Précis	Nanohybrid	Kerr
Amaris	Nanohybrid	VOCO
Admira Fusion	Nanohybrid	VOCO
Grandio	Nanohybrid	VOCO
GrandioSO	Nanohybrid	VOCO
Vittra	Nanofilled	FGM
Z350	Nanofilled	3M ESPE
Palfique LX5	Supra-nanofilled	Tokuyama

**Table 2 polymers-14-04020-t002:** Sample size (n = 10, following other studies [4,7,13]) according to the shade of RBC and the curing mode.

RBC Trademarks	Shade A2			Additional Curing	Effects Shade		
	VALO	Emitter a Fit	Emitter Now Duo	Emitter a Fit + Microwave	VALO	Emitter a Fit	Emitter Now Duo
	Group 1	Group 2	Group 3	Group 4	Group 5	Group 6	Group 7
Charisma	10	10	-	-	-	-	-
Z100	10	10	10	10	-	-	-
Z250	10	10	-	-	-	-	-
Polofil Supra	10	10	-	-	-	-	-
Empress Direct	10	10	-	-	10	10	10
Harmonize	10	10	10	10	-	-	-
Herculite Précis	10	10	-	-	10	10	10
Amaris	10	10	-	-	10	10	10
Admira Fusion	10	10	-	-	10	10	10
Grandio	10	10	10	10	20	20	20
GrandioSO	10	10	-	-	10	10	10
Vittra	10	10	10	10	10	10	10
Z350	10	10	10	10	10	10	-
Palfique LX5	10	10	-	-	10	10	10
TOTAL	140	140	50	50	100	100	90

“-” means that there were no specimens tested.

**Table 3 polymers-14-04020-t003:** Photopolymerization units, classification, and wavelength.

Photopolymerization Units	Classification	Wavelength
VALO	Third generation	395–480 nm
Emitter Now Duo	Third generation	385–515 nm
Emitter Fit A	Second generation	420–480 nm

**Table 4 polymers-14-04020-t004:** Mean and standard deviation of the biaxial flexure strength tests (in N) of each resin-based composite (shade A2) using LEDs from second- and third-generation curing units.

RBC Trademark	Emitter a Fit (1250 mW/cm^2^)	VALO (1100 mW/cm^2^)
Charisma A2	265.14 ^A 1,2^ (±51.13)	298.20 ^A 2,3^ (±75.33)
Palfique A2	175.07 ^A 1^ (±24.28)	168.63 ^A 1^ (±31.73)
Vittra A2	329.69 ^A 2,4^ (±59.75)	328.86 ^A 3^ (±85.45)
Empress A2	203.17 ^A 1,3^ (±40.03)	186.95 ^A 1,2^ (±54.38)
Z100 A2	307.04 ^A 2,3,4^ (±29.16)	363.85 ^A 3^ (±81.17)
Z250 A2	329.93 ^A 2,4^ (±126.23)	363.16 ^A 3^ (±81.17)
Z350 A2	375.28 ^A 2^ (±83.16)	355.32 ^A 3^ (±90.56)
Harmonize A2	368.53 ^A 2^ (±46.30)	348.88 ^A 3^ (±51.50)
Herculite Precis A2	268.12 ^A 1,2^ (±74.46)	301.68 ^A 2,3^ (±71.03)
Polofil Supra A2	357.81 ^A 2^ (±72.88)	287.21 ^A 1,3^ (±49.82)
Amaris O2	212.73 ^A 1,4^ (±39.55)	185.97 ^A 1,2^ (±30.97)
Admira Fusion A2	227.02 ^A 1,4^ (±53.09)	224.06 ^A 1,2,4^ (±31.14)
GrandioSO A2	379.65 ^A 2^ (±91.65)	299.53 ^A 2,3^ (±58.59)
Grandio A2	381.51 ^A 2^ (±53.34)	328.27 ^A 3,4^ (±70.17)

Different letters in the same row denote a statistically significant difference (*p* > 0.05). Different numbers in the same column denote a statistically significant difference (*p* > 0.05).

**Table 5 polymers-14-04020-t005:** Mean and standard deviation of the biaxial flexure strength test (in N) of each resin-based composite (shade A2) cured with a second-generation LED unit and microwave.

RBC Trademark	Emitter a Fit	Emitter a Fit + Microwave
Z350 A2	375.28 (±83.16)	334.99 (±50.19)
Z100 A2	307.04 (±29.14)	317.42 (±83.02)
Grandio A2	381.51 (±53.84)	301.08 (±76.42)
Harmonize A2	368.53 (±46.30)	346.81 (±49.38)
Vittra A2	329.69 (±59.75)	251.58 (±59.62)

No statistically significant differences were found within the same RBC and the same curing mode (*p* > 0.05).

**Table 6 polymers-14-04020-t006:** Mean and standard deviation of the biaxial flexure strength test (in Newtons (N)) of each resin-based composite (effects shade) cured with LED units from the second and third generations.

RBC Trademark	Emitter a Fit	VALO	Emitter Now Duo
Palfique CE	187.69 ^A 1^ (±32.11)	163.00 ^A 1,5^ (±26.76)	183.50 ^A 1^ (±46.70)
Z350 GT	365.93 ^A 2,3^ (±62.08)	385.28 ^A 2^ (±72.85)	-----
Herculite precis LTI	295.45^A1,2,3^ (±66.87)	297.55 ^A 2,3^ (±51.80)	308.10 ^A 2,3^ (±79.27)
Grandio Incisal	358.81 ^A 2,3^ (±52.58)	309.01 ^A 2,4^ (±33.96)	321.41 ^A 2^ (±111.78)
Empress BL-L	223.61 ^A 1,3^ (±50.46)	175.06 ^A 1,3^ (±66.43)	136.35 ^A 1^ (±42.99)
Amaris Translúcida	211.48 ^A 1,3^ (±50.81)	197.81 ^A 1,3,4^ (±34.14)	190.60 ^A 1,3^ (±53.80)
Admira fusion Incisal	241.63 ^A 3^ (±42.26)	207.76 ^A 1,3,4^ (±52.76)	153.87 ^A 1^ (±39.51)
Grandio BL	325.16 ^A 3^ (±63.08)	310.57 ^A 4,5,6^ (±60.64)	316.52 ^A 2^ (±109.39)
GrandioSO BL	315.55 ^A 3^ (±64.50)	292.31 ^A 2,3,4^ (±79.83)	308.31 ^A 2,3^ (±59.65)
Vittra E Bleach	310.73 ^A 1,3^ (±88.72)	394.57 ^A 2,6^ (±115.73)	306.97 ^A 2,3^ (±117.74)

Different letters in the same row denote a statistically significant difference (*p* > 0.05). Different numbers in the same column denote a statistically significant difference (*p* > 0.05). Shade effects mean that RBCs do not have hue and chroma.

**Table 7 polymers-14-04020-t007:** Mean and standard deviation of the biaxial flexure strength test (in N) of resin-based composites of different shades cured with second- and third-generation LED units.

RBC Trademark	Emitter a Fit		VALO		Emitter Now Duo	
	Shade A2	Effect Shade	Shade A2	Effect Shade	Shade A2	Effect Shade
Palfique	175.07 (±24.28)	187.69 (±32.11)	168.63 (±31.73)	163.00 (±26.76)	-----	183.5 (±46.70)
Z350	375.28 (±83.16)	365.93 (±62.08)	298.2 (±75.33)	385.28 (±72.85)	297.72 (±51.80)	-----
Herculite precis	268.12 (±74.46)	295.45 (±66.87)	186.95 (±54.38)	297.55 (±51.80)	-----	308.10 (±79.27)
Grandio	381.51 (±53.34)	358.81 (±52.58)	363.85 (±81.17)	309.01 (±33.96)	332.05 (±63.81)	321.41 (±111.78)
Empress	203.17 (±40.03)	223.61 (±50.46)	363.16 (±81.17)	175.06 (±66.43)	-----	136.35 (±42.99)
Amaris	212.73 (±39.55)	211.48 (±50.81)	355.32 (±90.56)	197.81 (±34.14)	-----	190.6 (±53.80)
Admisa fusion	227.02 (±53.09)	241.63 (±42.26)	348.88 (±51.50)	207.76 (±52.76)	-----	153.87 (±39.51)
GrandioSO	379.65 (±91.65)	315.55 (±64.80)	301.68 (±71.03)	292.31 (±79.83)	-----	308.31 (±59.65)
Vittra	329.69 (±59.75)	310.73 (±88.72)	287.21 (±49.82)	394.57 (±115.73)	251.58 (±59.62)	306.97 (±117.74)

No statistically significant differences were found between the RBC of shade A2 and shade effects within the same curing mode (*p* > 0.05). Shade effects mean that the RBCs do not have hue and chroma.

**Table 8 polymers-14-04020-t008:** Degree of conversion (%) of each RBC, according to the curing mode.

RBC Trademark	Emitter a Fit	VALO	Emitter Now Duo
Admira Fusion Incisal	53.1	54.2	51.6
Admira Fusion A2	53.4	51.5	54.4
Amaris Translúcido	37.8	38.7	37.3
Amaris O2	37.4	37.9	37.5
Charisma A2	43.1	47.9	43.3
Empress BL-L	30.4	32.6	30.3
Empress A2	38.1	34.3	30.9
Grandio Incisal	44.7	47.5	45.0
Grandio BL	43.3	43.3	41.8
Grandio A2	42.2	44.0	40.4
GransioSO BL	45.9	46.8	46.8
Grandioso A2	46.9	48.6	45.7
Harmonize A2	43.7	47.9	44.1
Herculite Précis A2	50.0	51.2	50.1
Palfique LX5 CE	34.8	35.3	37.1
Palfique A2	40.1	42.8	39.6
Polofil Supra A2	58.2	56.3	54.8
Vittra E Bleach	31.3	66.2	48.5
Vittra A2	42.8	50.0	35.9
Z100 A2	40.0	39.5	38.2
Z250 A2	49.3	48.3	47.4
Z350 GT	51.1	50.0	51.0
Z350 A2	44.6	50.6	46.1

Statistical analysis was not performed since only one specimen of each RBC was used.

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
