# Peer review of "Effect of Curing Modes on the Mechanical Properties of Commercial Dental Resin-Based Composites: Comparison between Different LEDs and Microwave Units"

_polymers, 2022, doi:10.3390/polym14194020_

Round 1
Reviewer 1 Report
Dear Authors
You can find the comments in the attached file.

Author Response
Thank you very much for spend your time to get my paper better.
My answers is attached.
Thank you

Reviewer 2 Report
The manuscript Effect of curing modes on the mechanical properties of commercial dental resin-based composites" shows many specimen tested. The manuscript needs more scientific information.
1. The reviewer suggest to include the chemical reaction in formula of one specimen what taking place during photo-activation and under what condition can reduce the formation.
2. The introduction needs as well some more explanation. The authors used a lot of samples and which of those most applied (some statistic). if possible include those
3. Making analytical studies the conditions how those tested should vary only in small content. How did the authors determine which are cured well and others cured less. Please include more of these details in the result parts.
4. Those RCB are applied in cavity filling and how does the condition inside the mouth place a role in curing? Did the authors applied those under same conditions?
5. It would be also interesting how does those RCB changed in view of characterization (FTIR etc) before and after activation. Are some of those studies made (a reference would be sufficient)
6. The authors used a lot of samples. What is the statistic are there some din't work (what is the failure rate?).
Author Response
Thank you very much for spend your time to get my paper better.
My answers is below.
Thank you
The manuscript Effect of curing modes on the mechanical properties of commercial dental resin-based composites" shows many specimen tested. The manuscript needs more scientific information.
Dear reviewer thank you very much to spend your time to get this work better.
- The reviewer suggest to include the chemical reaction in formula of one specimen what taking place during photo-activation and under what condition can reduce the formation.
It was included in the text as figure 1.
- The introduction needs as well some more explanation. The authors used a lot of samples and which of those most applied (some statistic). if possible include those
The introduction section was improved.
- Making analytical studies the conditions how those tested should vary only in small content. How did the authors determine which are cured well and others cured less. Please include more of these details in the result parts.
This information was included in result section.
- Those RCB are applied in cavity filling and how does the condition inside the mouth place a role in curing? Did the authors applied those under same conditions?
Yes, this study simulated different restorations as Venners, Class IV, Class V, Class III…, the next projects will be to study the influence of thickness of composite resin and distance of light source more related with cavities (class I and 2)
- It would be also interesting how does those RCB changed in view of characterization (FTIR etc) before and after activation. Are some of those studies made (a reference would be sufficient)
It was included as figure 3.
- The authors used a lot of samples. What is the statistic are there some din't work (what is the failure rate?).
Anova test was used to statistical analyses (p>.05). Tukey's test was used for multiple comparison at a significance level of 5% using the SPSS statistic software (IBM, New York, NY, USA). Both statical analysis worked well.
Round 2
Reviewer 1 Report
Dear Authors
Thank You for your time spent in response my comments. The topic of this article is very interesting and could be useful to dental clinician. However, some critical issues remained unsolved.
1) The title express “Effect of curing modes on the mechanical properties of commercial dental resin-based composites.” However in lines 112-115 Authors stated “VALO (light-curing for 15 seconds with an irradiance of 1100 mW/cm2); 112 Emitter A Fit (light-curing for 15 seconds with an irradiance of 1250 mW/cm2); Emitter 113 Now Duo (light-curing for 15 seconds with an irradiance of 1250 mW/cm2); and an additional curing with a microwave (BMS45, Brastemp) for 3 minutes at 450 W.”, then basically the article treats only 15 second curing and microwave curing. I suggest to clearly explain the tested curing modes.
2) Figure 1: I suggest putting only the formula in word format. Moreover, there are “+” inside white square in the figure. Please correct
3) Lines 68-70: Please Rephrased
4) Lines 108-115: “The RBC specimens were then divided into seven groups according to the curing 108 modes (Valo (Groups 1 and 5), Emitter A Fit (Groups 2 and 6), Emitter Now Duo (Groups 109 3 and 7)., and Emitter A fit + microwave) Group 4) and shades (A2 (Groups 1, 2, 3, and 4), 110 and effects shades (Groups 5, 6, and 7)) of the RBCs as described in Table 2” I suggest to rephrased with a list of the single groups. For example, “ RBC specimens were then divided into seven groups according to the curing modes:
Group 1:
Group 2:
…and so on,” specifying both the curing lamp and the curing modes.
5) Table 2: What does “-” mean? For example, Charisma Group 3? Please add this information in the table capture. Moreover, Authors should explain why all RBCs were not assessed with all curing lamp and protocols.
6) Lines 117-121: I suggest summarizing this information in a Table.
7) Lines 142-143: How were the readings performed? Which were bands considered in this study and why? Please explain in the text.
8) Figure 3: Please translate in English.
9) Lines 151-154: Since the numbers of the tested RBC, I suggest Authors describing the calculation of the sample size.
10) Lines 157-158: I suggest moving this sentence in the Material and Methods.
11) Table 3,4,5: Authors used +-.+/- to describe the standard deviation. Please use only one of that. Moreover, I suggest using ±
12) Lines 173-174: “Table 5 displays the comparisons of RBC of effects shade according to the curing 173 mode.”. In Table 5 there are only three different curing lamp and not curing mode.
13) Lines 173-174: “As much greater the degree of conversion, more polymerized was the composite 174 resin”. Authors should explain this sentence because on of the method to describe the polymerization process is degree of conversion.
14) Table 6: What does Effect Shade mean? Please explain in the text.
15) Lines 182-183: What does (in N) mean? Please explain.
16) Line 259: Explain what poor polymerization means, and which value do you consider as poor polymerization. Moreover Reference 28 does not treat this issue. Please rephrased.
17) Authors should explain the Discussion section the reason in using microwave and how it could be useful in clinical practice
Author Response
Dear Authors
Thank You for your time spent in response my comments. The topic of this article is very interesting and could be useful to dental clinician. However, some critical issues remained unsolved.
Thank you for help us to get better this paper.
- The title express “Effect of curing modes on the mechanical properties of commercial dental resin-based composites.” However in lines 112-115 Authors stated “VALO (light-curing for 15 seconds with an irradiance of 1100 mW/cm2); 112 Emitter A Fit (light-curing for 15 seconds with an irradiance of 1250 mW/cm2); Emitter 113 Now Duo (light-curing for 15 seconds with an irradiance of 1250 mW/cm2); and an additional curing with a microwave (BMS45, Brastemp) for 3 minutes at 450 W.”, then basically the article treats only 15 second curing and microwave curing. I suggest to clearly explain the tested curing modes.
Valo and emitter now duo is a 3 generation of photopolimerizator, they have blue, violet, and ultraviolet light that achieve the photoiniciators while Emitter fit is a 2 generation that has only a blue light. This was include in the text
- Figure 1: I suggest putting only the formula in word format. Moreover, there are “+” inside white square in the figure. Please correct
It was done as suggested
- Lines 68-70: Please Rephrased
It was done in the text
4) Lines 108-115: “The RBC specimens were then divided into seven groups according to the curing 108 modes (Valo (Groups 1 and 5), Emitter A Fit (Groups 2 and 6), Emitter Now Duo (Groups 109 3 and 7)., and Emitter A fit + microwave) Group 4) and shades (A2 (Groups 1, 2, 3, and 4), 110 and effects shades (Groups 5, 6, and 7)) of the RBCs as described in Table 2” I suggest to rephrased with a list of the single groups. For example, “ RBC specimens were then divided into seven groups according to the curing modes:
Group 1:
Group 2:
…and so on,” specifying both the curing lamp and the curing modes.
It was done as suggested
- Table 2: What does “-” mean? For example, Charisma Group 3? Please add this information in the table capture. Moreover, Authors should explain why all RBCs were not assessed with all curing lamp and protocols.
The informations was included in the table. For microwave and effects shades the authors would like just some examples and another reason is that some composite resins do not have effect shades.
- Lines 117-121: I suggest summarizing this information in a Table.
It was done as suggested
- Lines 142-143: How were the readings performed? Which were bands considered in this study and why? Please explain in the text.
All informations to explain this were included in the text
- Figure 3: Please translate in English.
It was done as suggested
- Lines 151-154: Since the numbers of the tested RBC, I suggest Authors describing the calculation of the sample size.
It was not done because it is already assumed in literature in this king of research that this number of specimens is enough to do the tests.
- Lines 157-158: I suggest moving this sentence in the Material and Methods.
It was done as suggested
- Table 3,4,5: Authors used +-.+/- to describe the standard deviation. Please use only one of that. Moreover, I suggest using ±
It was done as suggested
- Lines 173-174: “Table 5 displays the comparisons of RBC of effects shade according to the curing 173 mode.”. In Table 5 there are only three different curing lamp and not curing mode.
It was changed in the text.
- Lines 173-174: “As much greater the degree of conversion, more polymerized was the composite 174 resin”. Authors should explain this sentence because on of the method to describe the polymerization process is degree of conversion.
This was changed in te text
- Table 6: What does Effect Shade mean? Please explain in the text.
This was included in the text.
- Lines 182-183: What does (in N) mean? Please explain
It was included in the text.
- Line 259: Explain what poor polymerization means, and which value do you consider as poor polymerization. Moreover Reference 28 does not treat this issue. Please rephrased.
It was changed in the text
- Authors should explain the Discussion section the reason in using microwave and how it could be useful in clinical practice
It was done as suggested
Round 3
Reviewer 1 Report
Dear Authors
Thank you for the time spent in responding my comments. However some more minor aspect should be discussed in the text:
Abstract
Add the seconds used for all the curing modes.
Authors might consider changing the title “ Effect of curing modes on the mechanical properties of commercial dental resin-based composites: comparison between LED and Microwave unit”. This could make the title more relevant to the manuscript.
Materials and Methods
Since Authors decided on the sample size using other articles, I strongly recommend indicating this in the paper, adding the related reference.
Line 179: “Anova test was used to statistical analyses (p>.05).” p> or p< ?
Table 3: Please add also the power of the lamp.
Line 203-204: Please explain this sentence and add reference.
Discussion
I suggest also discussing the reason on using 15 seconds for the lamp and 3 minutes for the microwave, adding the related references.
Line 297-305: The Authors stated “There is also, as demonstrated in the literature, a possibility to increase the degree of 297 conversion of a composite resin, using a microwave. This would make a restoration, made 298 indirectly (veneer, onlay, inlay, or overlay), better compared to another one polymerized 299 directly in the mouth, thus increasing its resistance”. Since microwave is a method used outside the mouth for curing indirect restorations, other dental laboratory curing units might be tested instead LED units. I suggest Authors to add this consideration in the limitation of the study or as further study.
Author Response
Comments and Suggestions for Authors
Dear Authors
Thank you for the time spent in responding my comments. However some more minor aspect should be discussed in the text:
Thank you so much
Abstract
Add the seconds used for all the curing modes.
It was done as required.
Authors might consider changing the title “ Effect of curing modes on the mechanical properties of commercial dental resin-based composites: comparison between LED and Microwave unit”. This could make the title more relevant to the manuscript.
This suggestion was accepted.
Materials and Methods
Since Authors decided on the sample size using other articles, I strongly recommend indicating this in the paper, adding the related reference.
It was included in the text
Line 179: “Anova test was used to statistical analyses (p>.05).” p> or p< ?
It was corrected in the text.
Table 3: Please add also the power of the lamp.
It was done as suggested
Line 203-204: Please explain this sentence and add reference.
It was included in the text
Discussion
I suggest also discussing the reason on using 15 seconds for the lamp and 3 minutes for the microwave, adding the related references.
The microwave was just an additional energy to composite resin after LED, this information was included in the text.
Line 297-305: The Authors stated “There is also, as demonstrated in the literature, a possibility to increase the degree of 297 conversion of a composite resin, using a microwave. This would make a restoration, made 298 indirectly (veneer, onlay, inlay, or overlay), better compared to another one polymerized 299 directly in the mouth, thus increasing its resistance”. Since microwave is a method used outside the mouth for curing indirect restorations, other dental laboratory curing units might be tested instead LED units. I suggest Authors to add this consideration in the limitation of the study or as further study.
This was included in the test.
